# SNOOPPI: SEQUENCE-NORMALIZED DATABASE OF ON- AND OFF-TARGET PROTEIN-PROTEIN INTERACTIONS

**Sophia Vincoff**[1], **Pranam Chatterjee**[1,2,†]

[1]Department of Bioengineering, University of Pennsylvania
[2]Department of Computer and Information Science, University of Pennsylvania

[†]Corresponding author: pranam@seas.upenn.edu

## ABSTRACT

The set of physical protein-protein interactions (PPIs) realized in a cell defines a functional proteome whose interaction patterns constrain and characterize cellular state. PPIs are therefore central means by which biological processes are executed and therapeutic interventions act. Here, we introduce **SNOOPPI**, a **S**equence-**N**ormalized database of **O**n- and **O**ff-target **P**rotein-**P**rotein **I**nteractions, which represents the first unified dataset of binary PPIs that is isoform, post-translational modification, mutation, and binding site aware. By defining a PPI as a direct, physical interaction between two amino acid sequences, SNOOPPI overcomes several persistent limitations of existing PPI databases. SNOOPPI was curated from the IntAct database, taking full advantage of its experimental metadata and feature annotations to reclassify and uncover new PPIs. The final dataset comprises over 35.2K positive interactions and 5.3K negative interactions. SNOOPPI also retains 834.3K unresolved interactions, explicitly capturing gaps in the experimental literature. Beyond its usefulness as a reference dataset for the scientific community, SNOOPPI has the potential to serve as a high-confidence foundation for sequence-based modeling, benchmarking, and generative design of novel protein perturbations.

## 1 INTRODUCTION

Protein-protein interactions (PPIs) underlie fundamental biological processes in health and disease (Akbarzadeh et al., 2024; Greenblatt et al., 2024). Molecules that form or disrupt PPIs are central to biological investigation and therapeutic intervention across infectious disease, cancer, and neurodegeneration (Lu et al., 2020; Alfaris et al., 2024; Chan et al., 2025). Machine learning has accelerated both PPI network inference (Xiong et al., 2025a; Evans et al., 2021; Burke et al., 2023) and therapeutic molecule design (Chen et al., 2025a; Brixi et al., 2023; Bhat et al., 2025; Chen et al., 2025b; Tang et al., 2025a; Chen et al., 2025c; Vincoff et al., 2025a; Tang et al., 2025b; Chen et al., 2025d), yet progress in discovering, interpreting, and predicting PPIs remains fundamentally constrained by data quality (Tsishyn et al., 2024; Neumann et al., 2022).

Experimental PPI detection methods vary widely in throughput, reliability, and reported evidence. High-throughput assays such as yeast-two-hybrid and affinity purification mass spectrometry scale efficiently but exhibit elevated false-positive rates (Brückner et al., 2009; Gnanasekaran & Pappu, 2023), whereas lower-throughput approaches including nuclear magnetic resonance (NMR), X-ray crystallography, fluorescence resonance energy transfer (FRET), and surface plasmon resonance (SPR) provide higher confidence but limited coverage (Akbarzadeh et al., 2024; Peng et al., 2017). These assays produce heterogeneous outputs, ranging from binary interaction calls to kinetic parameters and atomic structures (Akbarzadeh et al., 2024). Each method is further constrained in the types of interactions it can detect. Co-complex approaches cannot resolve direct contacts within $n$-ary assemblies (Peng et al., 2017); heterologous binary assays may miss interactions dependent on native localization, PTMs, or cofactors; and structural methods struggle with weak interactions and intrinsically disordered proteins despite high spatial resolution (Akbarzadeh et al., 2024; Busch et al., 2025; Haubrich et al., 2023).

This heterogeneity forces PPI databases to make explicit choices about representation, confidence, and scope. Some resources prioritize functional associations, including STRING (Szklarczyk et al., 2025), Reactome (Milacic et al., 2024), and KEGG (Kanehisa et al., 2023). Others aim to catalog physical interactions through literature curation and integration, including IMEx Consortium databases (Del Toro et al., 2022; Xenarios et al., 2002; Zanzoni et al., 2002; Samarasinghe et al., 2025; Kotlyar et al., 2025; Breuer et al., 2013) and BioGRID (Oughtred et al., 2021), which adopt the PSI-MI XML standard (Bader et al., 2003). Structural repositories such as the RCSB PDB (Berman et al., 2000) capture experimentally resolved complexes, but require additional filtering and augmentation to infer biologically relevant PPIs (Liu et al., 2015; Zhang et al., 2024; Wei et al., 2024; Kovtun et al., 2024; Bushuiev et al., 2023; Chen et al., 2025b; Bhat et al., 2025). Similar challenges persist across other interaction resources (Zhu et al., 2025; Veres et al., 2015; Du et al., 2021). Confidence scoring schemes partially address these issues (Del Toro et al., 2022; Alanis-Lobato et al., 2016), yet most databases remain overwhelmingly positive-only due to the rarity of explicitly reported non-interactions, with limited exceptions derived from literature and structure-based analyses (Del Toro et al., 2022; Blohm et al., 2014; Kovtun et al., 2024).

Despite these efforts, core problems remain unresolved. Protein identities are inconsistently specified, negative evidence is rarely explicit, and assay failure is often indistinguishable from absence of interaction. Structure-based non-interactions can be particularly misleading, as lack of a resolved interface does not preclude interaction *in vivo* (Wei et al., 2024; Zhang et al., 2024). Consequently, negative datasets are frequently constructed using heuristic criteria such as random pairing, separation by localization or function, or exclusion by homology, introducing biases that simplify prediction tasks rather than reflect biological constraints (Neumann et al., 2022). Although PSI-MI provides a rich annotation framework, extracting sequence-specific, binary, direct interactions from resources such as IntAct remains difficult without substantial computational effort (Del Toro et al., 2022).

To address all of these limitations, we introduce a comprehensively **S**equence-**N**ormalized database of **O**n- and **O**ff-target **P**rotein-**P**rotein **I**nteractions, termed **SNOOPPI**. SNOOPPI is motivated by the observation that proteins are ultimately defined by their exact amino acid sequences, including isoforms, mutations, and PTMs, yet this information is often abstracted away in existing PPI resources. Evidence for interaction loss, arising from targeted mutations, PTM perturbations, or binding-site deletions, is frequently embedded within detailed annotations but not surfaced as explicit negative evidence. As such, we present a systematic reorganization of IntAct into a PPI and PepPI database indexed by the precise sequences of both binding partners. SNOOPPI focuses on direct, physical, binary interactions and explicitly records the experimental evidence supporting interaction and non-interaction claims. To demonstrate SNOOPPI's natural extension to sequence-based PPI classification, we provide homology-disjoint data splits and train baseline models to serve as benchmarks for the field. By centering sequence identity and exposing evidentiary provenance, SNOOPPI provides a reproducible and interpretable foundation for biological analysis and language-model-based modeling of protein interactions.

## 2 MATERIALS AND METHODS

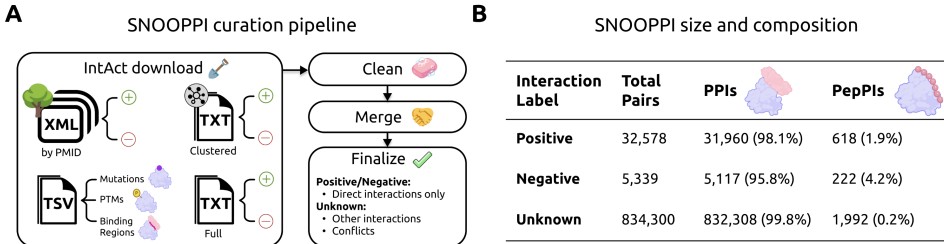

Figure 1: **SNOOPPI database of direct, binary, sequence-based PPIs. A** Overview of the IntAct download and reorganization process. **B** SNOOPPI database composition, including positive, negative, and unknown PPIs and PepPIs. A peptide is defined as containing at most 50 amino acids.

### 2.1 SOURCE DATA

All source data were derived from IntAct (Del Toro et al., 2022) Release 251 (September 2025). IntAct was selected because it uniquely combines controlled vocabularies (PSI-MI 3.0 (Sivade et al.,

2018)), explicit amino acid sequences for most interactions, isoform, PTM, mutation, and binding-region awareness, inclusion of PPIs involving intrinsically disordered or structurally intractable proteins, literature-derived non-interacting pairs, clear distinction between directly detected binary interactions and those inferred via matrix or spoke expansion, comprehensive molecular interaction (MI) confidence scores, integration with other PPI databases, and frequent updates. Together, these properties are required for constructing a reliable, sequence-resolved PPI dataset. Several non-IMEx resources were considered (Oughtred et al., 2021; Szklarczyk et al., 2025; Peri et al., 2004; Zhu et al., 2025), but each lacked one or more of these criteria.

Although IntAct provides a user-friendly web interface and complete downloads, its data are fragmented across XML, MITAB, TXT, and TSV formats, making it difficult to obtain a unified, sequence-based PPI dataset. Amino acid sequences are stored only in XML files, while interaction expansion information appears exclusively in specific TXT files. At the same time, the small number of reported negative PPIs (894 in intact-micluster_negative.txt) contrasts sharply with the large number of feature annotations, including 205,575 binding-region entries, 10,283 PTMs, and 83,626 mutations, suggesting that many experimentally supported non-interactions are implicitly encoded but not surfaced. Mutated sequences are not provided directly by IntAct and instead appear as feature annotations specifying coordinates and residue changes.

To resolve these issues, we reprocessed the raw XML files corresponding to individual PPI detection experiments and harmonized them with the aggregated TXT and TSV files (Figure 1A). This pipeline yielded 32,578 positive and 5,339 negative PPIs that are direct, binary interactions with known sequences for both partners. Interactions with missing sequences, conflicting or insufficient evidence, or interaction designations weaker than MI:0407 (direct interaction) were reassigned to SNOOPPI-unknown, which contains 834,300 PPIs (Figure 1B). Most interactions are protein-protein, although each dataset includes hundreds of peptide-protein interactions, with a small number of peptide-peptide interactions among the positive and unknown sets.

## 2.2 Defining a protein-protein interaction

The term "protein-protein interaction" has been used to describe colocalization, functional association, complex membership, and intramolecular contacts (Akbarzadeh et al., 2024). Here, we define a PPI strictly as a direct, physical contact between two (poly)peptide chains in an experimental setting. Because this definition operates at the level of a physical chain rather than a gene or transcript, the exact amino acid sequence of each partner is required, including any PTMs. A protein sequence is therefore treated as text encoding the full molecular state at the time of evidence collection, such that modified and unmodified residues are represented distinctly. The identity of a PPI is defined exclusively as the order-agnostic pair of the two interacting sequences.

This binary definition permits homomeric interactions, excludes strictly intramolecular interactions, and excludes complexes containing more than two distinct amino acid sequences, while retaining binary PPIs with non-1:1 stoichiometries. Interactions may be transient or stable and may span a wide range of affinities. A peptide-protein interaction (PepPI) is defined identically, except that one partner must be shorter than 50 amino acids. This cutoff follows prior work (Chen et al., 2025a) but required reclassification of several IntAct entries.

A "negative PPI" denotes a pair of sequences that do not form a direct, binary, physical interaction based on experimental evidence. IntAct's raw negatives are manually curated from studies reporting lack of interaction, with most direct negatives identified via pull-down assays (PSI-MI MI:0096). We do not filter interactions by assay type, but retain assay metadata and IntAct MI-scores, which incorporate assay-dependent evidence weighting. Most negative PPIs in SNOOPPI are newly derived from mutation and binding-region annotations. For mutation-derived sequences, we can guarantee experimental testing. For necessary binding-region annotations (MI:0429), the PSI-MI hierarchy indicates that deletion or mutation abolishes interaction, but the exact tested sequence variants are not provided by IntAct, and experimental testing of those precise sequences cannot be guaranteed.

An "unknown PPI" refers to any binary (poly)peptide sequence pair present in IntAct that either lacks decisive evidence of being positive or negative under our definition, or contains conflicting evidence supporting both. SNOOPPI Unknown does not enumerate untested protein pairs, but captures cases where IntAct provides binary PPI or PepPI evidence that remains inconclusive.

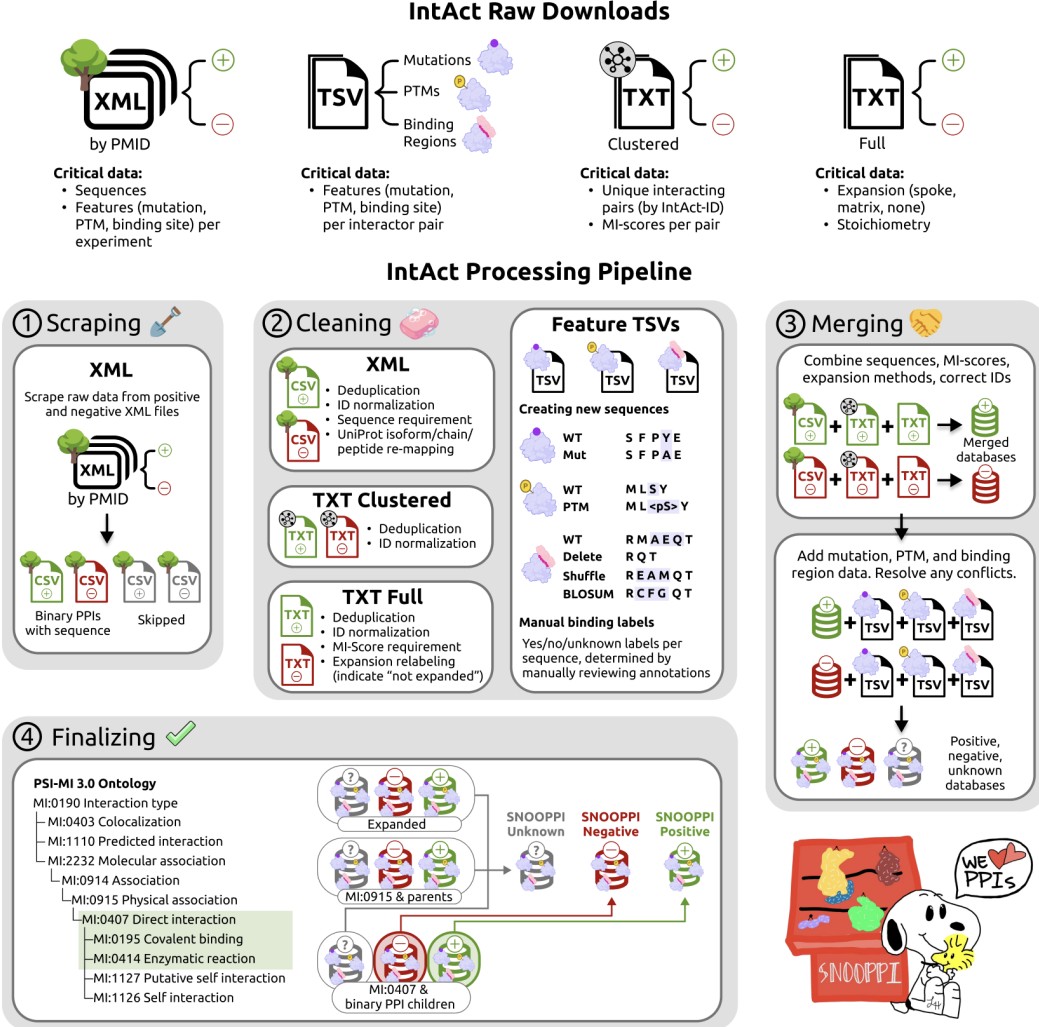

Figure 2: **SNOOPPI full data curation pipeline.**

## 2.3 PROCESSING AND FILTERING INTACT

We provide step-by-step documentation of the SNOOPPI curation pipeline in Appendix A and ensure full reproducibility through our open-source codebase. Figure 2 summarizes the major processing stages and their objectives.

**IntAct Raw Downloads**

Four IntAct file types were downloaded: (1) XML files organized by year and PubMed ID, (2) feature TSVs describing mutations, PTMs, and binding regions, (3) clustered TXT files where each row corresponds to a unique PPI, and (4) non-clustered ("full") TXT files where each row represents a single piece of experimental evidence. Separate files were provided for positive and negative PPIs in both XML and TXT formats.

XML files are hierarchical element trees that store maximal experimental detail unavailable through the IntAct web interface or other formats. Critically, XML files contain the precise amino acid sequences used in each experiment, derived from the original publication. These sequences can be mapped to specific UniProt isoforms, chains, propeptides, transit peptides, signal peptides, and other subsequences. XML files also include per-experiment feature annotations, enabling identification of mutations, PTMs, and binding regions that affect interactions but were tested only in $n$-ary ($n > 2$) or expanded experiments.

The feature TSVs provide interpretable summaries of mutation, PTM, and binding-region annotations. While fully recoverable from the XML files, these TSVs are resolved only at the interaction and interactor identifier level. Because multiple experiments can contribute to a single IntAct interaction, TSVs alone do not specify which features correspond to binary, non-expanded evidence. We therefore used the TSVs to cross-check XML-derived data and restrict analysis to relevant interactions.

The clustered TXT files define a unique interaction as the order-agnostic pairing of two IntAct identifiers and provide a single MI-score per interaction. The full TXT files retain much of the XML detail and are the only format that explicitly records whether spoke or matrix expansion was applied. Expansion status is not indicated in XML files, and clustered TXTs do not distinguish interactions supported by non-expanded evidence.

**IntAct Processing Pipeline**

XML data were extracted via programmatic scraping (Figure 2, part 1). Interactions involving more than two partners or any non-(poly)peptide participants were excluded (Table A1). All datasets were then cleaned (Figure 2, part 2) through deduplication and identifier normalization. Only XML entries containing sequences for both partners were retained. UniProt ID Mapping was applied to obtain updated UniProt accessions, including isoform, chain, and subsequence identifiers. In the full TXT files, experiments lacking expansion annotations were explicitly labeled as "not expanded."

TSV processing focused on reconstructing full partner sequences and determining binding outcomes. Mutation and PTM TSVs typically specify both original and modified residues, allowing direct reconstruction of full-length sequences. Because no standardized PTM sequence encoding exists, PTMs were represented using the full PSI-MI term supplied by IntAct. For example, a tyrosine at position 1 ("Y") could be encoded as <Y1+psi-mod:"MOD:00181(O4'-sulfo-L-tyrosine)">, with brackets denoting a nonstandard token. Binding-region TSVs identify regions necessary for interaction (MI:0429) but do not specify the exact tested sequences. Based on the MI:0429 definition, we generated three non-interacting variants per sequence: deletion of the binding region, random shuffling of the region, and residue-wise substitution with the least likely BLOSUM62 substitution. The shuffled control reflects common experimental practice (Wu et al., 2016), while the BLOSUM62-based variant (Henikoff & Henikoff, 1992) minimizes chemical similarity to maximize disruption likelihood. Binding outcomes were determined by manual review of all MI terms and feature annotations associated with mutations, PTMs, and binding regions. Each annotation was labeled as "yes", "no", or "unknown" (MI-term categorizations in Tables A2, A2, and A4). Labels were applied at the TSV row level and then aggregated by amino acid sequence to assign final interaction labels. Additional cleaning details are provided in Appendix A.

Cleaned datasets were merged (Figure 2, part 3), and conflicts necessitated creation of an Unknown category. For example, when mutation and PTM annotations could not be confirmed as co-occurring, both derived sequences were assigned to Unknown. Sequence pairs with conflicting experimental evidence, showing both interaction and non-interaction, were also assigned to Unknown.

Finally, merged data were filtered by interaction type (Figure 2, part 4). Only MI:0407 ("direct interaction") and its subtrees rooted at MI:0195 ("covalent binding") and MI:0414 ("enzymatic reaction") were accepted as evidence of direct physical interaction. MI:1227 ("putative self interaction") entries were reclassified as Unknown due to ambiguity between homomeric and intramolecular interactions. MI:1226 ("self interaction") entries were excluded entirely as confirmed unary interactions. The final SNOOPPI Unknown dataset aggregates expanded interactions, interactions labeled with MI:1227 or parents of MI:0407, and all other interactions failing decisive classification. Only direct positive and negative interactions were retained in the final SNOOPPI Positive and SNOOPPI Negative datasets.

## 2.4 ESTABLISHING A PPI CLASSIFICATION BASELINE

SNOOPPI was converted into rigorous train, validation, and test splits, which are applicable to any machine learning PPI classifier. Here, we train a suite of baseline models.

**Data Splitting Pipeline**

We began by determining criteria for a comprehensive, minimally biased, and maximally challenging data split: (1) high over all ratio of negatives to positives (RNP), reflecting the real-world sparsity of PPIs, (2) equal RNP for each protein, preventing bias towards interaction "hubs", (3) no data leakage: no sequence >30% homologous to *either* protein in a given pair may appear in a different

split, (4) test set with as many gold standard PPIs (MI-score >= 0.8) as possible, (5) presence of closely connected positive-negative PPI pairs (proteins are mutants/isoforms of each other, or differ only in the binding region) in all splits. To enable homology analyses, all SNOOPPI proteins were clustered using MMSeqs2 (Steinegger & Söding, 2017) at 30%, 50%, and 70% homology with 80% coverage.

Criteria 1-2 required the addition of random negatives to bolster total and per-protein RNP. For each protein in SNOOPPI Positive, random SNOOPPI proteins at least 30% homologous to one known partner, but no more than 70% homologous to any known partner, were identified as candidates. A greedy algorithm was used to simultaneously select and split optimal PPI/negative PPI pairs while minimizing computational cost. The algorithm capped per-protein positives, enforced an approximate RNP and train/val/test ratio, prioritized real and hard negatives over randoms, prioritized difficult randoms (more homologous to a true partner) over easy ones, preserved closely connected positive-negative bundles, biased gold standard groups towards the test set, and, most critically, guaranteed complete absence of homologous sequence leakage.

**Model Architectures and Training**

Based on prior work, we selected a set of classical ML models and deep neural network architectures as baselines Zhang et al. (2026). Additionally, two ESM-2 variants (the smallest, ESM-2-8M, and SOTA ESM-2-650M) and two standard encodings (one-hot and VHSE) were tested as sequence encoders, both on a per-token and length-pooled basis. Random Forest, XGBoostoost, and Elastic Net, as well as pooled CNN and MLP architectures, were trained on fixed-dimension embeddings. Four two-tower architectures, each featuring either CNN, MLP, Transformer, or Cross-Attention modules, were tested on unpooled embeddings. For each architecture, five Optuna (Akiba et al., 2019) trials were run for a brief hyperparameter optimization (Table C1). Models were chosen based on maximum F1 score.

## 3 RESULTS

### 3.1 SNOOPPI IS A SEQUENCE-BASED BINARY INTERACTOME

SNOOPPI is a sequence-indexed subset of IntAct. IntAct contains 1,098,988 positive PPIs, of which 37,488 are direct, and 894 negative PPIs, of which 37 are direct, with each PPI represented as a row in the clustered XML files (Del Toro et al., 2022). IntAct further provides 10,283 PTM, 83,626 mutation, and 205,575 binding-region annotations (Figure 3A, left). Using these annotations and interaction-type MI terms, SNOOPPI labels 834,300 interactions as Unknown (Figure 3A, right) and identifies 32,578 positive and 5,339 negative PPIs that are direct, binary, non-expanded, and sequence-resolved. PTM annotations added or confirmed 325 interactions and reassigned 1,325 to Unknown. Mutation annotations substantially expanded the dataset, adding 3,492 high-confidence negatives, confirming or adding 6,632 positives, and reclassifying 1,426 Unknowns. Binding-region features added 1,827 negative PPIs; however, because many MI terms do not establish necessity (Table A4), the corresponding mutants were assigned to Unknown. Finally, interaction-type MI terms alone reclassified 692,928 interactions as Unknown (Figure 3A, right).

SNOOPPI-Positive spans 666 species across all domains of life (Figure 3B). Eukaryotes dominate, comprising 9,381 sequences (82%), with mammals contributing the largest share alongside hundreds of fungal, plant, and other animal proteins. Positive PPI partners also include bacteria (1,244), Archaea (86), and viruses (554), and 214 peptides were chemically synthesized (Figure 3B, left). The ten most represented species or sources, in descending order, are human, yeast, mouse, *A. thaliana*, *E. coli*, rat, *D. melanogaster*, chemical synthesis, *C. elegans*, and SARS-CoV-2. Corresponding analyses for SNOOPPI-Negative and SNOOPPI-Unknown are shown in Figure B1.

Protein and peptide lengths span a broad range (Figure 3C). Values are shown up to the 97th percentile, with outliers extending to 35,000 amino acids. Across databases, average protein length is approximately 500 amino acids, with an interquartile range of roughly 250–800 amino acids. Peptides in the positive and negative sets are slightly shorter on average (12–13 amino acids) than those in the Unknown set (mean 24 amino acids).

MI-scores, which summarize interaction confidence based on evidence quantity and quality, are skewed toward intermediate values across all datasets, with highest density between 0.4 and 0.6 (Figure 3D). High-confidence interactions with MI-scores between 0.8 and 1.0 include 2,451 positives (7.5%) and 970 negatives (18.0%), defining the most confident subset of SNOOPPI (Figure 3D).

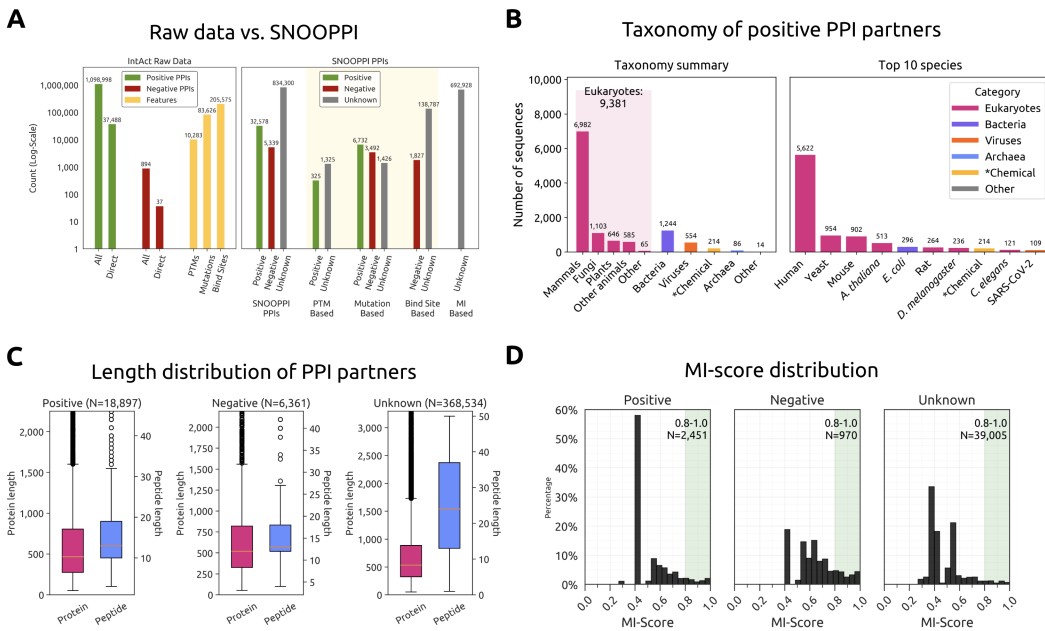

Figure 3: **SNOOPPI data composition. A** IntAct provides thousands of annotations that can augment its positive and negative PPI datasets (left). SNOOPPI uses PTM, mutation, and binding site annotations, as well as interaction MI-terms, to confirm or reclassify IntAct's PPIs (right). **B** Taxonomic (left) and species (right) distributions of the individual proteins participating in SNOOPPI's positive PPIs. *Chemical represents chemical synthesis, typically referring to a peptide. **C** Protein and peptide sequence lengths in SNOOPPI Positive (left), Negative (middle), and Unknown (right). Values up to the 97th percentile are displayed. **D** Distribution of MI-scores across SNOOPPI Positive (left), Negative (middle), and Unknown (right). The total number of interactions with high MI-scores (0.8-1.0) is displayed for each dataset.

## 3.2 SNOOPPI ENABLES A SEQUENCE-BASED PPI CLASSIFICATION BENCHMARK

SNOOPPI is a natural training dataset for purely sequence-based binary PPI classification. There are several valid approaches to data splitting; here, we provide a strict cluster-disjoint split, which significantly shrinks dataset size but rigorously evaluates model generalizability. Across the training (48,828 pairs; 81%), validation (5,093 pairs; 8.5%), and test (6,306 pairs; 10.5%) splits, no partner shares more than 30% sequence homology with any partner in a different split (Table 1). Our greedy pairing and splitting algorithm established a fairly consistent ratio of negatives to positives (between 9-10), maximized retention of real negatives over random, and prioritized placement of highest-quality pairs (MI-score >= 0.8) in the test set (Table 1).

Table 1: Statistics of the SNOOPPI dataset after pairing, selection, and homology-aware splitting.

| Split | Total | Pos | Neg | Neg/Pos | Gold Std | Real Neg | Unique Proteins |
|-------|-------|-----|-----|---------|----------|----------|-----------------|
| Train | 48,828 | 4,470 | 44,358 | 9.92 | 0.0% | 1,411 | 6,910 |
| Val | 5,093 | 505 | 4,588 | 9.09 | 2.3% | 187 | 794 |
| Test | 6,306 | 601 | 5,705 | 9.49 | 13.3% | 621 | 1,395 |

We trained nine baseline model architectures on various pooled ($\mathbf{x} \in \mathbb{R}^d$) and unpooled ($\mathbf{x} \in \mathbb{R}^{L \times d}$) sequence representations. For most architectures, ESM-2 (Lin et al., 2023) embeddings consistently improved performance compared to one-hot and VHSE. The two tower CNN architecture prevailed with F1 score of 0.40 (Table 2). Full results are provided in Table C1.

Table 2: Best results for each baseline classifier model

| Model | Pooling | Embedding | Best F1 Score |
|---|---|---|---|
| Elastic Net | Pooled | One Hot | 0.185 |
| Random Forest | Pooled | ESM-2-8M | 0.245 |
| XGBoost | Pooled | ESM-2-650M | 0.267 |
| CNN Pooled | Pooled | ESM-2-8M | 0.185 |
| MLP Pooled | Pooled | ESM-2-650M | 0.265 |
| **Two Tower CNN Unpooled** | **Unpooled** | **ESM-2-650M** | **0.400** |
| Two Tower MLP Unpooled | Unpooled | ESM-2-650M | 0.222 |
| Two Tower Transformer Unpooled | Unpooled | ESM-2-650M | 0.249 |
| Two Tower Cross-Attention Unpooled | Unpooled | ESM-2-8M | 0.215 |

## 4 DISCUSSION

Modeling cellular state and environments has increasingly focused on transcriptomic, epigenetic, and spatial representations of state, yet cellular behavior is ultimately executed through physical protein-protein interactions (PPIs). In this sense, the PPI network is one of the most defining and mechanistically grounded representations of a cellular state. SNOOPPI is the first sequence-normalized dataset of experimentally verified PPIs and PepPIs, spanning positive, negative, and explicitly unresolved interactions. Although several resources provide amino acid sequences (Del Toro et al., 2022; Zhang et al., 2024; Chen et al., 2025a; Zhu et al., 2025), they are typically organized around publications, experiments, or protein identifiers (uni, 2025; Brown et al., 2015; Berman et al., 2000) rather than indexed by the exact sequences that were assayed, making it difficult to determine whether two specific sequences physically interact. PTMs, mutations, and isoform differences are often treated as annotations rather than first-class entities, obscuring their functional effects. SNOOPPI adopts a sequence-first representation, defining each interaction as a binary, direct physical contact between two precisely defined sequences, with mutations and PTMs incorporated directly, reducing ambiguity and resolving interactions beyond the gene level.

A central motivation for SNOOPPI is its utility for machine learning and sequence-based design. Many deep learning models for PPI prediction (Ko et al., 2024; Singh et al., 2022), PepPI prediction (Abdin et al., 2022; Xiong et al., 2025b; Bhat et al., 2025), and interface identification (Wang et al., 2025; Abdin et al., 2022; Xiong et al., 2025b) are limited by training data assembled indirectly from identifiers, structures, or heterogeneous assays, with negative datasets often encoding strong and unexamined assumptions (Neumann et al., 2022). By explicitly representing positive, negative, and unresolved interactions at the sequence level, SNOOPPI provides a consistent starting point for prediction. Our trained baseline models constitute a fair benchmark for binary PPI classification, their modest F1 scores reflecting the difficulty of the task. SNOOPPI is also well-positioned to support generative or optimization-based design of novel interactors (Brixi et al., 2023; Bhat et al., 2025; Chen et al., 2025a;b; Vincoff et al., 2025a), with natural extensions to antibodies, nanobodies, non-canonical amino acids, and chemically modified peptides via SMILES representations (Tang et al., 2025a; Chen et al., 2025c; Tang et al., 2025b; Chen et al., 2025d). The dataset has limitations, including a large unresolved set, evolving ontologies, incomplete PTM standardization (Peng et al., 2025), bias toward alanine mutations, exclusion of multimeric complexes, and underrepresentation of rare but clinically relevant classes such as fusion oncoproteins (Vincoff et al., 2025b). Even with these constraints, SNOOPPI provides a precise, evidence-grounded, sequence-resolved foundation for modeling physical protein interactions as a core layer of cellular state.

## DECLARATIONS

## 5 MEANINGFULNESS STATEMENT

Cellular state is ultimately realized through physical protein–protein interactions (PPIs), which determine signaling, regulation, and response to perturbation. SNOOPPI provides a sequence-resolved, experimentally grounded representation of this interaction layer by explicitly encoding positive, negative, and unresolved PPIs at the level of assayed amino acid sequences, including isoforms, mutations, and post-translational modifications. By surfacing interaction and non-interaction evidence

that is typically implicit or lost, SNOOPPI reduces reliance on heuristic negatives and enables more faithful training and evaluation of sequence-based models. In the context of virtual cell modeling, SNOOPPI offers a concrete data foundation for learning how cellular state is defined and altered through physical molecular interactions.

## 6 DATA AVAILABILITY

The full database can be accessed and downloaded through an interactive HuggingFace page, https://huggingface.co/datasets/ChatterjeeLab/SNOOPPI. All code is open-source and maintained on GitHub at https://github.com/sophievincoff/interactome. The database and related code will be updated every six months.

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

APPENDIX

## A  DATA CURATION

### A.1  INTACT PROCESSING

The IntAct database provides a centralized resource for literature-derived, expert-verified PPIs across a wide variety of experimental methods and organisms. IntAct aggregates all data contributed by members of the IMEx Consortium, who have agreed to follow common curation standards. Active database members as of November 2025 include: IntAct (Del Toro et al., 2022), DIP (Xenarios et al., 2002), MINT (Zanzoni et al., 2002), MatrixDB (Samarasinghe et al., 2025), IID (Kotlyar et al., 2025), InnateDB (Breuer et al., 2013), UniProt (uni, 2025), and EMBL-EBI (Thakur et al., 2024). Inactive members (contributed in the past) include: databases MPact (Güldener et al., 2006), BIND (Bader et al., 2003), MPIDB (Goll et al., 2008), Molecular Connections, MBInfo, HPIDB (Ammari et al., 2016), and the UCL-BHF group. As a starting point for our binary PPI dataset, IntAct Release 251 -

September 2025 was downloaded. The full download presents IntAct's data in several formats. We utilize (1) PSI30-formatted (Sivade et al., 2018) XML files organized by year and PubMed ID, (2) TXT files organized by experiment, (3) TXT files clustered by interacting pair, and (4) three separate TSV files describing binding regions, mutations, and PTMs that impact PPIs.

Each file format contains largely overlapping but sometimes distinct content. This ultimately necessitated the full processing and recombination of all XML, TXT, and TSV files. Below, we outline each file format with a description, download information, and list of unique contributions relative to other file types. All file paths are relative to https://ftp.ebi.ac.uk/pub/databases/intact/current.

**1. XML files** Found at: psi30/pmid, in subfolders named for the year in which the article was published. Unique contributions:

1. **Amino acid sequences.** The precise amino acid sequences of each protein or peptide involved in the interaction.

2. **Feature annotations (PTM, mutation, and binding site) per experiment.** In the aggregated TSV files (format 4) for each of these feature types, it is not possible to identify the exact experiment associated with the feature. This is problematic when the same study provides both binary and *n*-ary interactions involving the same pair of proteins, and a PTM or mutation was only tested in the *n*-ary experiment. We limit our curation to binary interactions and their associated features only.

**2. TXT files indexed by experiment** Found at: psimitab/intact.txt (positive PPIs) and psimitab/intact_negative.txt (negative PPIs). Unique contributions:

1. **Expansion labels per experiment.** The raw XML files do not indicate whether a binary PPI was determined via matrix expansion, spoke expansion, or no expansion. Format (3) (clustered TXT files) aggregates expansion labels but excludes no-expansion; *e.g.*, if an interaction was determined once with no expansion and once with spoke expansion, its entry under "Expansion method(s)" would only read: psi-mi:"MI:1060"(spoke expansion). An interaction that has been verified without expansion is more likely to be a true binary PPI. Therefore, our pipeline reassigns more comprehensive labels to each pair, indicating all expansion modes including none.

**3. TXT files clustered by interacting pair**. Found at: psimitab/intact-micluster.txt (positive PPIs) and psimitab/intact-micluster_negative.txt (negative PPIs). These files are consolidated versions of intact.txt and intact_negative.txt, clustered by the pair of interactors. IntAct assigns a unique ID in the format "intact:EBI-*N*" to each unique combination of amino acid sequence and primary database accession. The primary database can be any IMEx Consortium member besides IntAct, such as UniProt, Ensembl, or DIP. For example, two proteins with distinct UniProt accessions that have identical sequences will receive separate intact:EBI identifiers; two proteins with the same UniProt accession but different sequences (*e.g.* isoforms) will receive separate intact:EBI identifiers. One unique pair of interacting proteins refers to the set of the two partners' intact:EBI IDs (in any order). IntAct clusters on these pairs in order to assign a single confidence score to each PPI. Unique contributions:

1. **Unique PPI identification**. This file confirms that each unique combination of two intact:EBI IDs represents a distinct PPI, helping to guide our aggregation of data scraped from the raw XML.

2. **One interaction score per pair**. The clustered file provides a single "mi-score" which encapsulates all experiments that support the interaction of two distinct proteins, as represented by their intact:EBI IDs.

4. **Feature TSV files**: Found at psimitab/features/ptms.tsv, psimitab/features/mutations.tsv, and psimitab/features/binding_regions.tsv. These files present PTM, mutation, and binding region data for each experiment. They do not provide unique data relative to the other files (all of these annotations can be scraped directly from XML) but were helpful for cross-checking to confirm the correctness of the pipeline.

In addition to these files, the controlled vocabularies at cv/intact.obo were used to categorize MI terms relating to interaction type (*e.g.* MI:0407: direct interaction, MI:0573: mutation disrupting interaction).

A.1.1 SCRAPING THE RAW XML

Most of the raw data stored in the XML files was extracted in our scraping process. This includes:

1. Interaction-level information:

   (a) Interaction label (*e.g.* "direct interaction")

   (b) Interaction MI (*e.g.* "MI:0407")

   (c) Interaction IntAct ID (*e.g.* "intact:EBI-16189444")

   (d) Experiments (including PubMed ID, method (*e.g.* "anti bait coip"), hosts with taxonomic ID (*e.g.* "9606") and short/full labels (*e.g.* "human", "Homo sapiens"))

   (e) Year

   (f) Processing method ("xml" or "pymex")

   It should be noted that the `mif` module of the `pymex` package unexpectedly failed to parse many of the XML entries. To prevent data loss, we performed all parsing with `xml.etree.ElementTree`, and thus all processing methods are "xml".

2. Interactor-level information:

   (a) Gene name (*e.g.* "rpb1_human")

   (b) Gene symbol (*e.g.* "POLR2A")

   (c) Molecule type ("protein" or "peptide")

   (d) Species label (*e.g.* "human")

   (e) Species taxonomic ID (*e.g.* "9606")

   (f) Amino acid sequence (*e.g.* "MHGGGPP...")

   (g) Length of amino acid sequence

   (h) Chain start and end coordinates for the provided sequence (1-indexed and end position-inclusive, *e.g.* "1-405")

   (i) All UniProt, ENSP, ENSG, ENST, RCSB PDB, IntAct, and DIP accessions

   (j) All InterPro, Reactome, and GO annotations

   (k) Name and ID of primary database (*e.g.* "uniprotkb", "Q16637")

   (l) Host information: taxonomic ID, full label, short label, cell type (*e.g.* "293t" for HEK293T cells) cellular compartment (*e.g.* "nucleus"), tissue (*e.g.* "colon")

   (m) Mutation annotations:

       i. MI (*e.g.* "MI:0573")

       ii. Name (*e.g.* "mutation disrupting interaction")

       iii. Short name (*e.g.* "E9M5R0:p.[Asp293_Ser294delinsAsnAla]")

       iv. Beginning coordinates (*e.g.* "293,296") and ending coordinates (*e.g.* "294,297")

       v. Original sequence (*e.g.* "DS,SE") and new sequence (*e.g.* "NA,AQ") at the specified coordinates

       Sometimes, multiple mutations were applied at once. In these cases, the beginning/ending coordinates and original/new sequences were stored in comma-separated format (*e.g.* begin: "1,3", end: "1,3", original: "M,P", new: "A,A"). There were also many XML entries where multiple mutation experiments were reported. These were pipe-separated (*e.g.* begin: "1,3|60,61", end: "1,3|60,61", original: "M,P|R,R", new: "A,A|G,G") This curation strategy provided a clear distinction between mutations that were applied together and those that were not. Note that all coordinates are all

1-indexed and inclusive, meaning that if the beginning and ending coordinates are both "1", then the mutation was only applied to the first amino acid in the sequence.

(n) PTM annotations:

  i. MI (*e.g.* "MI:1224")

  ii. Name (*e.g.* "increasing-ptm,observed-ptm,N6-acetyl-L-lysine")

  iii. Short name (*e.g.* "possible acetyllys-310")

  iv. Beginning coordinates (*e.g.* "310") and ending coordinates (*e.g.* "310")

  v. Original and new sequences (rarely provided)

When multiple positions had PTMs or multiple PTM features were reported in one XML block, the same protocol was followed as for mutations.

(o) Binding region annotations:

  i. MI (*e.g.* "MI:0429")

  ii. Name (*e.g.* "necessary binding region")

  iii. Short name (*e.g.* "binding region")

  iv. Beginning coordinates (*e.g.* "186") and ending coordinates (*e.g.* "292")

Positive and negative files were processed separately. Only entries that contained exactly two protein participants were included in the output file; any *n*-ary interactions were excluded. Additionally, interactions that did not strictly involve proteins and/or peptides were excluded. All skipped interactions were saved in a separate file with columns denoting the following: file name, interaction XML ID, IntAct ID for the interaction, and the reason. The interaction XML ID was pulled out of a block formatted according to the following example:

```
<interaction id="6" imexId="IM-23272-1">
```

The XML ID (in this example, 6) was stored because in the same file, the same two proteins could be shown to interact in different XML interaction blocks, but only some blocks may hold valid binary PPIs.

This process produced four output files: "intact_processed_positivePPIs.csv", "intact_processed_negativePPIs.csv", and "intact_skipped_positivePPIs.csv". No negative interactions were skipped (Table A1).

Table A1: Interactions scraped from IntAct

| Type | Accept/Reject | Total | Reason |
|---|---|---|---|
| Positive | Accept | 746,032 | |
| Positive | Reject | 69,681 | not binary |
| Positive | Reject | 55,022 | not PPI or PepPI |
| Negative | Accept | 969 | |

A.1.2 CLEANING THE SCRAPED XML DATA AND TXT FILES

Here, we outline each step taken to clean the scraped XML data and the raw TXT files downloaded from IntAct. Initially, database sizes were as follows: intact.txt = 1,726,476, intact-micluster.txt = 1,136,486, intact_negative.txt = 984, intact-micluster_negative.txt = 931, scraped file intact_processed_positivePPIs.csv = 746,032, scraped file intact_processed_negativePPIs.csv = 969.

**Cleaning the non-clustered TXT files**

To clean "intact.txt" and "intact_negative.txt", the following key steps were taken:

1. Deduplication (deleted copies of identical rows)

2. Dropped any row lacking an IntAct ID for Interactor A or Interactor B

3. Dropped any rows lacking an mi-score

4. Created a designation for binary PPIs that were truly detected as binary, and not inferred by spoke or matrix expansion. In all rows where ""Expansion method(s)" was missing, we created a label: "not expanded". This had a significant impact, leaving the database with 916,419 rows (53.12%) labeled as "psi-mi:"MI:1060"(spoke expansion)" and 808,646 rows (46.88%) labeled as "not expanded."

5. Exploded along the column "IntAct Interaction identifier(s)" so that each row corresponded to just one intact:EBI ID for the interaction. Only three rows from "intact.txt" and zero from "intact_negative.txt" originally had multiple IntAct interaction identifiers.

**Cleaning the micluster files**

To clean "intact-micluster.txt" and "intact-micluster_negative.txt", the following key steps were taken:

1. Deduplication (deleted copies of identical rows)

2. Dropped any row lacking an IntAct ID for Interactor A or Interactor B

3. Confirmed that the unique combination of intact:EBI identifiers for Interactor A and Interactor B only appears once throughout the database

**Cleaning the scraped XML files**

To clean "intact_processed_positivePPIs.csv" and "intact_processed_negativePPIs.csv", the following key steps were taken. We note that in these scraped files, Interactors A and B are instead referred to as Interactors 1 and 2. In the merging process, this difference in notation clarified which columns came from the scraped XML and which came from the raw IntAct TXT downloads. To minimize confusion, we will continue to use the terminology A and B in this section.

1. Deduplication (deleted copies of identical rows)

2. Dropped any row lacking an amino acid sequence for Interactor A or Interactor B

3. Checked that each row has at least one IntAct identifier for Interactors A and B.

4. Exploded along the columns containing these identifiers. The resulting database contained exactly one IntAct ID for each interactor in each row.

5. Exploded along the column containing the IntAct identifiers for the interaction as a whole. The resulting database contained exactly one IntAct ID representing the interaction in each row.

6. Prepared for merging by flipping (and effectively doubling) the database so we could later match it to the corresponding "intact-micluster(_negative)" rows even if the interactors were in the opposite order.

### A.1.3 MERGING THE SCRAPED XML DATA WITH TXT FILES

Here, we outline every step taken to merge the scraped XML data with the raw TXT downloads from IntAct, after cleaning. Information in "intact_processed_positivePPIs.csv" was combined with the raw downloads "intact.txt" and "intact-micluster.txt". Information in "intact_processed_negativePPIs.csv" was combined with the raw downloads "intact_negative.txt" and "intact-micluster_negative.txt". In the text below, any mention of grouping or investigating "by PPI" refers to the order-agnostic set of the IntAct IDs of Interactor A and B.

1. Grouped "intact.txt" by PPI and created a mapping between each PPI and all its detected mi-scores and expansion methods.

2. Verified that any mi-score detected for a PPI in "intact(_negative).txt" matched the aggregated mi-score for that PPI in "intact-micluster(_negative).txt"

3. Utilized the mapping from part (1) to assign every row in "intact-micluster.txt" a full set of relevant expansion methods.

4. Filtered "intact-micluster(_negative).txt" to retain only rows where at least one expansion method was "not expanded". This resulted in 485,913/1,136,283 (42.7%) of positive rows and 921/931 (98.9%) of negative rows.

5. Exploded "intact-micluster(_negative).txt" along the column containing interaction IntAct IDs. The resulting database was no longer indexed by PPI and had several rows with the same PPIs and different interaction IDs (representing different studies and different experiments detecting the same PPI).

6. Merged the cleaned XML data with the appropriate "intact-micluster(_negative)" data on the following: interaction IntAct ID, PPI, interactor A IntAct ID, interactor B IntAct ID. This ensured that only one configuration of Interactor A-Interactor B was retained per PPI per interaction IntAct ID.

7. **UniProt ID Remapping**: The unique list of all UniProt IDs for Interactors A and B was collected. Rather than collecting the specific isoform provided (*e.g.* P35240-1), we trimmed the ID down to its canonical form (*e.g.* P35240). This list was submitted to the UniProt ID Mapping tool, under UniProt release 2025_04. Two results files were downloaded: (1) FASTA with canonical and isoform sequences, and (2) TSV with all default fields as well as Chain, Peptide, Propeptide, Signal Peptide, and Transit Peptide. These fields were added because many of IntAct's provided UniProt IDs ended with "-PRO...", indicating that they represented a chain or peptide. The chain, peptide, propeptide, signal peptide, and transit peptide sequences for each UniProt accession were extracted and added to the candidate list of sequences provided by the FASTA file. Each Interactor A and Interactor B sequence was re-matched to its correct UniProt ID with isoform/chain/peptide. In any case where the canonical isoform matched and it did not have a corresponding isoform (*e.g.* Q10173), a stand-in isoform "-0" was appended (*.e.g* Q10173-0), to indicate clearly that isoforms were considered for every sequence. All canonical UniProt IDs provided by IntAct were correct, unless they were deprecated (which was the case for about 2% each of interactors A and B).

These steps produced two intermediate files which represent the merging of XML, full TXT, and clustered TXT-derived data. Positive interactions are stored in "merged.txt" and negative in "merged_neg.txt"

### A.1.4    INCORPORATING MUTATIONS

The mutation annotations provided in the raw XML and "mutations.tsv" created an opportunity to extract additional positive and negative sequences. Each mutation feature is associated with an MI term, which was used to determine whether the original sequence binds its partner and whether the mutated sequence binds its partner.

According to the IntAct curation manual and Molecular Ontologies, a mutation is described as "disrupting" an interaction if and only if the interaction is completely abolished. Meanwhile, a mutation is described as "causing" an interaction if and only if the interaction cannot occur without the mutation. Using these definitions, we assume that "decreasing" and "increasing" imply the presence of an interaction both before and after mutation (Table A2).

Table A2: Categorization of mutation-related MI terms.

| MI | Definition | Original Binds | Mutated Binds |
|---|---|:---:|:---:|
| MI:2226 | mutation with no effect | yes | yes |
| MI:0382 | mutation increasing interaction | yes | yes |
| MI:1132 | mutation increasing interaction strength | yes | yes |
| MI:1131 | mutation increasing interaction rate | yes | yes |
| MI:0119 | mutation decreasing interaction | yes | yes |
| MI:1133 | mutation decreasing interaction strength | yes | yes |
| MI:1130 | mutation decreasing interaction rate | yes | yes |
| MI:0573 | mutation disrupting interaction | yes | no |
| MI:1128 | mutation disrupting interaction strength | yes | no |
| MI:1129 | mutation disrupting interaction rate | yes | no |
| MI:2227 | mutation causing an interaction | no | yes |
| MI:0118 | mutation | unknown | unknown |
| MI:2333 | mutation with complex effect | unknown | unknown |

The following steps were taken to incorporate mutation data into "merged.txt" (positive PPIs and PepPIs) and "merged_neg.txt" (negative PPIs and PepPIs).

1. Combined TSV and XML-derived data.

    (a) Exploded "merged" and "merged_neg" by interaction IntAct ID, so that each row contained exactly one interaction identifier.

    (b) Identified columns from XML-scraping that relate to mutations: "mutation_mi_*N*", "mutation_name_*N*", "mutation_short_*N*", "mutation_begin_*N*", "mutation_end_*N*", "mutation_orig_*N*", "mutation_new_*N*", where *N* references either interactor "A" or "B". Exploded both databases by the columns which relate to interactor A; then by the columns which relate to interactor B. This action separated mutations which were collected from the same experiment XML block, but were actually applied separately. At this point, each distinct set of mutations resides in its own row.

    (c) Separated any row where both interactor A and B had mutation-related data into two rows: one per interactor + mutation. This decision reflects an assumption that no experiment mutated both partners at the same time. (We later made an exception for homomeric interactions and assumed that in these cases, both partners were always mutated at the same time).

    (d) Cleaned original/mutated sequence data scraped from XML such that spaces and escape sequences ("\r", "\n") were removed, *e.g.* cleaning "QQQ\r\nQQQQ" to "QQQQQQQ".

    (e) Merged the aggregated mutation data from "mutations.tsv" with the positive and negative databases cleaned in previous steps. Merging was based on matching interaction IntAct IDs.

    (f) Dropped any rows labeled with MI:0429, "necessary binding region". Binding regions will be incorporated in a separate processing pipeline, instead of being treated as mutations.

    (g) Dropped any row with no mutation data (from either XML-scraping, or the mutations TSV).

    (h) In each row, determined which partner ("A" or "B") was affected by mutation. This was done by matching the "Affected protein AC" column from the mutations TSV with the interactors' database accessions, which were previously collected from XML and TXT files. The "Affected protein AC" column utilized either a UniProt, DIP, or IntAct identifier to indicate which partner was mutated.

    (i) Cleaned up the formatting of starting and ending coordinates. Many coordinates were formatted like this example: "123..123-456..456". Such entries were simplified to "123-456".

    (j) Filtered the current merged databases to correct, high-quality mutation annotations. To do this, we ensured that one of the following criteria were met:

        i. Mutation data was scraped from XML, and not available in the TSV file. (It is unclear why these cases were missed in the TSV file. We manually confirmed that several of them were correctly pulled from the raw XML). OR,

        ii. Mutation data was available both in the XML and TSV file. In these cases, we were able to determine which partner was mutated (step 1h). Additionally, both the mutation range (*e.g.* 319-319) and short label (*e.g.* P12612:p.Arg319Ala) exactly matched between the XML-derived data and the TSV-derived data. Consistency between these critical fields confirmed that the XML-derived data had been scraped correctly.

    (k) Filled in mutation data for both partners when the PPI is a homomer. As stated in step 1c, this decision reflects an assumption that for homomeric interactions, both partners are always mutated.

    (l) The prior step necessitated recalculating which partners were affected by mutation. For rows with TSV-derived data, we once again looked for matches with "Affected protein

AC". For rows without TSV-derived data, determined if partner A, B, or both (only in homomeric interactions) were mutated.

(m) Regrouped on columns related to "Feature # AC", which do not meaningfully separate different mutation features. This fixed the problem of multiple "Feature # AC"s being assigned to mutation annotations that were otherwise the same.

(n) Dropped rows lacking amino acid sequences before and after mutation, within the specified coordinates.

(o) **Determined mutated sequences.** Utilized the XML-derived "aa_$N$", "mutation_begin_$N$", "mutation_end_$N$", "mutation_orig_$N$", and "mutation_new_$N$" columns to determine the full-length sequence of the protein after each mutation.

2. Assigned yes/no/unknown PPI labels to each pair involving a wildtype or mutated partner.

(a) **Manually labeled mutation-related features.** In addition to MI terms (Table A2), 895 feature annotations from the "Feature annotation(s)" column and 12 feature accessions from the "# Feature AC" column were analyzed for evidence of binding in either the original or mutated sequence.

(b) Merged the manually-labeled features with the rest of the mutation data. Feature type labels were merged on the "Feature type" column; Feature AC labels on "# Feature AC"; Feature annotations on "Feature annotation(s)".

(c) Determined whether each wildtype/mutated sequence forms a PPI using our manual labels - on a per-row basis. At this point, we had manual yes/no/unknown labels, for each wildtype *and* mutated sequence, indicating whether that sequence binds its partner in that row. We had three separate columns containing such labels, derived from three pieces of information: MI-term (*e.g.* MI:226), feature annotation (*e.g.* "comment:Mutation does not disrupt interaction"), and feature accession, which often was truly an annotation (*e.g.* "The mutant shows no significant binding activity"). In one column, we computed the unique set of binding designations per row (*e.g.* if MI-term = "yes", annotation = "yes", and feature annotation = "no", the unique set would be "yes,no"). Then, in another column, we computed a final decision. If "yes" and "no" were both present, the final label would be "unknown". Otherwise, "yes" or "no" would each dominate a co-occurring label of "unknown".

(d) **Determined whether each wildtype/mutated seuqence forms a PPI using our manual labels - on a per-*sequence* basis.** In this crucial step, we aggregated evidence from different experiments pertaining to the same two sequences. We found several cases of disagreement between rows that described the same sequence pair, *e.g.* one "yes" and one "no". In all of these cases, the disagreement could be clearly traced back to conflicting MI-terms. For example, the same mutation could be reported as "causing" and "disrupting" an interaction. These conflicts cannot be resolved, and therefore they were assigned final labels of "unknown".

3. **Created mutation-aware positive, negative, and unknown PPI databases.**

(a) Incorporated the mutation annotations into the larger PPI database, by matching on the following fields: interaction IntAct ID, order-agnostic combination of IntAct IDs for interactors A and B, and order-agnostic combination of *wildtype* sequences for interactors A and B. Effectively, we found rows in the "merged.txt" and "merged_neg" that pertained to the same study and combination of interactors, and we created separate rows for the original wildtype PPI, as well as each combination of wildtype and mutated sequence with evidence.

(b) Asserted that all sequences were valid, meaning they only contained the canonical twenty amino acids or U (selenocysteine).

(c) Calculated one "mutation_short" label for each row. For heteromers, this label was equal to "mutation_short_$N$" for the affected partner. For homomers, we pipe-joined the "mutation_short" labels for each interactor. For rows with no mutatation annotations, the label was set equal to np.nan (effectively, None).

(d) Reorganized such that the databases were indexed on the following fields: (1) interaction IntAct ID, (2) order-agnostic combination of IntAct IDs for interactors A and B, (3) order-agnostic combination of *wildtype* sequences for interactors A and B, and (4) mutation-short label.

We note that none of the intermediate databases were sequence-indexed. We kept the databases organized by piece of evidence throughout mutation, PTM, binding site, and interaction MI-term processing. Only at the final processing step were the three databases rearranged such that each row contained a unique pair of sequences.

### A.1.5   INCORPORATING PTMs

To ensure that all amino acid sequences accurately represent the state of each binding partner, we also incorporated PTM MIs and annotations. Many PTM-related MI and MOD terms denote the type of modification (*e.g.* psi-mi:"MI:0170"(phosphorylated residue)), which does not directly communicate binding or lack thereof. We instead utilized these terms to construct PTM-inclusive sequences.

For the categorizations below, we assume that "prerequisite-ptm" means the interaction will not occur without the PTM; "resulting-ptm" means the PTM was not present when the interaction was initiated; and terms such as "increasing" or "decreasing" imply a change in interaction strength rather than interaction gain or loss (Table A3). These assumptions are based on the IntAct curation manual definitions of MI:0925 ("observed-ptm") and its children. Additionally, several entries in the "Feature annotation(s)" column of "ptms.txt" utilized very similar vocabulary to the MI-terms. We label both PTM-related MI terms and similar annotations in Table A3.

Table A3: Categorization of PTM-related MI-terms and feature annotations.

| MI | Definition / Term | Wildtype Binds | PTM'd Binds |
|---|---|:---:|:---:|
| **Terms with PSI-MI identifiers** | | | |
| MI:0925 | observed-ptm | unknown | unknown |
| MI:0638 | prerequisite-ptm | no | yes |
| MI:0639 | resulting-ptm | yes | unknown |
| MI:1233 | resulting-cleavage | yes | unknown |
| MI:1223 | ptm decreasing an interaction | yes | yes |
| MI:1224 | ptm increasing an interaction | yes | yes |
| MI:1225 | ptm disrupting an interaction | yes | no |
| **Additional annotation terms (no PSI-MI ID)** | | | |
| | phosphorylation increasing strength | yes | yes |
| | observed-ptm:Resulting PTM | yes | unknown |
| | observed-ptm:Prerequisite-PTM | no | yes |

The processing steps for incorporating PTMs were mirrored those taken for mutations. Before finalizing the PTM-aware SNOOPPI database, we checked for rows that had conflicting mutation and PTM annotations and moved these instances to unknown.

### A.1.6   INCORPORATING BINDING SITES

Based on the definitions of binding site-related MI terms, only one term - "MI:0429: necessary binding region" - can be used to derive negative PPIs. The full description of MI:0429 in the OLS Ontology Search resource reads: "A sequence range within a molecule identified as being absolutely required for an interaction. The sequence may or may not be in direct physical contact with the interaction partner." For the other three terms, there is only evidence that binding *does* occur when the region is present; there is no evidence that binding ceases to occur when the region is mutated or deleted. Our categorizations of binding site-related MI terms can be found in Table A4.

## B   SNOOPPI COMPOSITION ADDITIONAL DETAILS

Table A4: Categorization of binding site-related MI terms.

| MI | Definition | Original: Binds | Removed Binding Site: Binds |
|---|---|---|---|
| MI:0429 | necessary binding region | yes | no |
| MI:0442 | sufficient binding region | yes | unknown |
| MI:0117 | binding-associated region | yes | unknown |
| MI:1125 | direct binding region | yes | unknown |

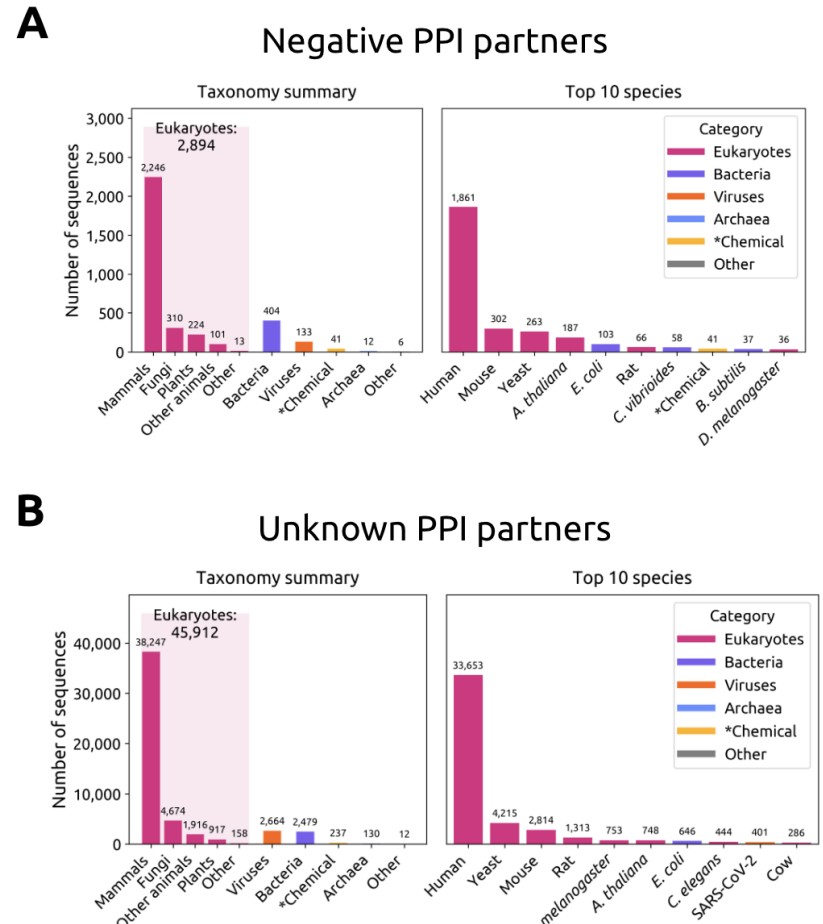

Figure B1: **Taxonomic distribution of SNOOPPI Negative and SNOOPPI Unknown PPI partners.** Higher-order taxonomic groups (left) and top ten species/sources for **A** SNOOPPI-Negative and **B** SNOOPPI-Unknown. *Chemical represents chemical synthesis, typically referring to a peptide

## C  BASELINE CLASSIFIERS ADDITIONAL DETAILS

Table C1: Full results of Optuna hyperparameter search on baseline architecture. Results are organized by model architecture and embedding type.

| Model | Pooling | Embedding | Best F1 |
|---|---|---|---|
| Elastic Net | Pooled | ESM-2-650M | 0.181 |
| Elastic Net | Pooled | ESM-2-8M | 0.182 |
| Elastic Net | Pooled | One Hot | 0.185 |
| Elastic Net | Pooled | VHSE | 0.184 |
| Random Forest | Pooled | ESM-2-650M | 0.239 |
| Random Forest | Pooled | ESM-2-8M | 0.245 |
| Random Forest | Pooled | One Hot | 0.220 |
| Random Forest | Pooled | VHSE | 0.200 |
| XGBoost | Pooled | ESM-2-650M | 0.267 |
| XGBoost | Pooled | ESM-2-8M | 0.259 |
| XGBoost | Pooled | One Hot | 0.202 |
| XGBoost | Pooled | VHSE | 0.205 |
| CNN Pooled | Pooled | ESM-2-650M | 0.183 |
| CNN Pooled | Pooled | ESM-2-8M | 0.185 |
| CNN Pooled | Pooled | One Hot | 0.183 |
| MLP Pooled | Pooled | ESM-2-650M | 0.265 |
| MLP Pooled | Pooled | ESM-2-8M | 0.255 |
| MLP Pooled | Pooled | One Hot | 0.190 |
| **Two Tower CNN Unpooled** | **Unpooled** | **ESM-2-650M** | **0.400** |
| Two Tower CNN Unpooled | Unpooled | ESM-2-8M | 0.233 |
| Two Tower CNN Unpooled | Unpooled | One Hot | 0.185 |
| Two Tower MLP Unpooled | Unpooled | ESM-2-650M | 0.222 |
| Two Tower MLP Unpooled | Unpooled | ESM-2-8M | 0.186 |
| Two Tower MLP Unpooled | Unpooled | One Hot | 0.185 |
| Two Tower Transformer Unpooled | Unpooled | ESM-2-650M | 0.249 |
| Two Tower Transformer Unpooled | Unpooled | ESM-2-8M | 0.247 |
| Two Tower Cross-Attention Unpooled | Unpooled | ESM-2-8M | 0.215 |

