# OpenReview forum: "SNOOPPI: Sequence-Normalized Database of On- and Off-Target Protein-Protein Interactions"
_ICLR.cc/2026/Workshop/FM4Science — ICLR 2026 Workshop FM4Science Poster_

### Official Review · Reviewer_eq9L · 2026-02-21
**Review on SNOOPPI: Sequence-Normalized Database of On- and Off-Target Protein-Protein Interactions**

**Rating:** 6
**Confidence:** 4

**Review:**

# Quality

The data pipeline is a strong technical aspect of this work, but the ML evaluation is limited. Only five Optuna trials per architecture is insufficient to draw reliable conclusions about architecture rankings. No ablation or error analysis is provided to distinguish dataset noise from model inadequacy.

# Clarity

The paper is generally well-written and well-organized with clear motivation in Section 1. Figure 1 provides an effective high-level overview, and Figure 2 comprehensively maps the curation pipeline.

# Originality

Redefining PPIs at the amino acid sequence level rather than the gene leve is a meaningful conceptual contribution. But APID (Alonso-López et al., 2019) already reprocesses IntAct with standardized PSI-MI evidence classification. While APID does not provide sequence-level indexing or explicit negatives, the approach of reprocessing and reclassifying IntAct data using the PSI-MI to extract binary interactions is not entirely new. The ML architectures and embeddings are standard, and the contribution here is primarily the dataset itself.

# Significance

SNOOPPI addresses a genuine gap in the PPI data landscape. The impact is somewhat limited by the small dataset size (35.2K positives, 5.3K negatives) relative to existing resources like STRING (~billions of associations) or BioGRID (~2M interactions).

# Pros

- SNOOPPI has principled sequence-first PPI definition that resolves persistent ambiguity in existing databases. It is a well-motivated design decision that enables isoform- and mutation-aware modeling.

- Systematic extraction of negative PPIs from mutation and binding-region annotations extends the tradition of experimentally grounded negative PPI datasets with a novel annotation-mining approach.

# Cons

- The composition of the negative set is not clearly disaggregated. The 5,339 experimental negatives are mixed with random negatives during splitting (Table 1), meaning ~96% of negatives in the benchmark are randomly generated.

- The PTM sequence encoding (Section 2.3) is introduced but not evaluated.

- The authors should clearly articulate how SNOOPPI's approach differs from previous databases.

---

### Official Review · Reviewer_b5Yp · 2026-02-22
**Although lack of structural context and experimental validation, SNOOPPI a thoughtfully curated sequence-resolved PPI resource,**

**Rating:** 10
**Confidence:** 5

**Review:**

This paper introduces SNOOPPI, a sequence-normalized database of direct, binary protein-protein interactions curated from IntAct. The authors re-organize existing interaction data at the level of exact amino acid sequences, explicitly incorporating isoforms, mutations, post-translational modifications (PTMs), and binding-site annotations. Importantly, the dataset includes not only positive interactions (32,578) but also 5,339 negative interactions and a large set of unresolved cases, addressing a major limitation of existing PPI resources that are overwhelmingly positive-only. I particularly appreciate that the authors systematically reclassified interactions using detailed experimental metadata, surfacing negative evidence that is often implicit in mutation and binding-region annotations. The explicit distinction between high-confidence negative interactions and the vast "unresolved" space is a commendable step toward reducing label noise in PPI modeling. This makes SNOOPPI, to my knowledge, the first sequence-resolved dataset that explicitly integrates both positive and negative PPIs while being mutation- and PTM-aware. The homology-disjoint splits and baseline models further establish a benchmark for purely sequence-based PPI prediction.
That said, there are several limitations. While sequence-based PPI prediction is appealing and scalable, amino acid sequence alone does not necessarily capture the physical conformation or biochemical properties of a PPI interface in the folded, tertiary (or quaternary) context. This limitation becomes even more pronounced when PTMs are involved, especially if they alter local structure, disorder, or binding geometry. Therefore, sequence-level annotation may not fully reflect the structural determinants of interaction, but this is a great starting point to address much more complex conditions. Finally, experimental validation of newly predicted positive and negative interactions derived from SNOOPPI is encouraged. Demonstrating that the dataset enables discovery of biologically verified novel PPIs (or confirmed non-interactions) would significantly strengthen its practical impact.
Overall, this is a carefully curated and conceptually important resource. I find the dataset construction thoughtful and valuable, though future work integrating structural context and experimental validation would enhance its translational relevance.

---

### Meta-Review · Area_Chair_4VbU · 2026-02-28

**Recommendation:** Accept (Poster)
**Confidence:** 4

**Metareview:**

This paper introduces SNOOPPI, a sequence-resolved protein–protein interaction database that explicitly incorporates isoforms, mutations, PTMs, positive interactions, high-confidence negatives, and unresolved cases. By redefining PPIs at the amino acid sequence level and mining experimental metadata for negative evidence, SNOOPPI addresses persistent ambiguity and positive-only bias in existing PPI resources. Reviewers find the dataset construction conceptually important and carefully executed. While the ML benchmarking component is limited and certain design choices (e.g., negative composition, PTM encoding evaluation) require clarification, the primary contribution — a principled sequence-level PPI resource — is strong and impactful.

---

### Decision · Program_Chairs · 2026-03-03

Accept (Poster)